# Nailfold Capillaroscopy: A Comprehensive Review on Its Usefulness in Both Clinical Diagnosis and Improving Unhealthy Dietary Lifestyles

**DOI:** 10.3390/nu16121914

**Published:** 2024-06-18

**Authors:** Michio Komai, Dan Takeno, Chiharu Fujii, Joe Nakano, Yusuke Ohsaki, Hitoshi Shirakawa

**Affiliations:** 1Laboratory of Nutrition, Graduate School of Agricultural Science, Tohoku University, Sendai 980-8572, Japan; yusuke.ohsaki.a4@tohoku.ac.jp (Y.O.); shirakah@tohoku.ac.jp (H.S.); 2At Co., Ltd., Osaka 541-0042, Japan; takenodan@kekkan-bijin.jp (D.T.); fujii@kekkan-bijin.jp (C.F.); jnakano@kekkan-bijin.jp (J.N.)

**Keywords:** nailfold capillaroscopy (NFC), medical examination, capillary morphology, dietary lifestyle, health claims, microcirculation, dietary flavonoids

## Abstract

Since the 1970s, the utility of nailfold capillaroscopy (NFC) in diagnosing rheumatological disorders such as systemic sclerosis has been well established. Further studies have also shown that NFC can detect non-rheumatic diseases such as diabetes, glaucoma, dermatitis, and Alzheimer disease. In the past decade, nailfold capillary morphological changes have also been reported as symptoms of unhealthy lifestyle habits such as poor diet, smoking, sleep deprivation, and even psychological stress, all of which contribute to slow blood flow. Therefore, studying the relationships between the morphology of nailfold capillaries and lifestyle habits has a high potential to indicate unhealthy states or even pre-disease conditions. Simple, inexpensive, and non-invasive methods such as NFC are important and useful for routine medical examinations. The present study began with a systematic literature search of the PubMed database followed by a summary of studies reporting the assessment of morphological changes detected by NFC, and a comprehensive review of NFC’s utility in clinical diagnosis and improving unhealthy dietary lifestyles. It culminates in a summary of dietary and lifestyle health promotion strategy, assessed based on NFC and other related measurements that indicate healthy microvascular blood flow and endothelial function.

## 1. Introduction

Nailfold capillaroscopy (NFC) is a simple, highly sensitive, non-invasive, safe, reliable, and inexpensive tool designed to assess a number of features of nailfold capillaries, including capillary density, blood flow, and different kinds of abnormal morphology. It can provide a visual inspection of the microcirculation in situ. Since the 1970s, NFC has gained recognition in helping to detect diseases that affect the microcirculation, especially many rheumatological disorders, such as connective tissue diseases (CTD) and rheumatic diseases like systemic sclerosis (SSc), rheumatoid arthritis (RA), systemic lupus erythematosus (SLE), and inflammatory myositis. Poor lifestyle and dietary habits can cause cardiovascular diseases, Alzheimer’s disease, diabetes, cancer, and other lifestyle-related diseases. A healthy lifestyle, including moderate exercise and the reducing the impact of negative factors such as psychological stress, smoking, and alcoholic intake, can prevent these diseases. Full-length articles published from 1980 to 2023, retrieved from PubMed using the search phrase “nailfold capillaroscopy”, are represented in Figure 1 [1].

The 2013 American College of Rheumatology (ACR)/European League Against Rheumatism (EULAR) classification criteria for SSc include two (out of the minimum of nine) points towards pathology classification for NFC SSc-specific findings in 2013 ([2], 2013). The inclusion of the NFC evaluation in the SSc classification criteria was based on its role in the early diagnosis of SSc; later the indications for capillaroscopy expanded [3,4,5,6]. Also, standardization guidelines for capillaroscopy practice for Raynaud’s phenomenon (RP) and SSc were recently published by the EULAR Study Group on Microcirculation in Rheumatic Diseases and the Scleroderma Clinical Trials Consortium on Capillaroscopy ([1], 2020).

The usefulness of NFC in the diagnosis of SSc is well known, and in other rheumatic conditions, such as various connective tissue diseases (e.g., vasculitis) and arthritis, nailfold video capillaroscopy (NVC) anomalies are often indicative of a scleroderma-like pattern. However, the utility of NFC in non-rheumatic diseases (NRDs) has not been well studied (for a review over the period of 1990 to 2019, see [7]). Microvascular damage detectable by capillaroscopy is caused by a number of other diseases, including diabetes, glaucoma, dermatology, sickle cell disease (SCD), interstitial lung disease (ILD), idiopathic pulmonary hypertension (i-PAH), Alzheimer’s disease (AD), and Rett syndrome (RS). NFC enables measurements to be evaluated for individual capillaries (length, shape, and diameter of each capillary loop, and the number of capillaries), hemorrhage, the extent of avascular area, angiogenesis, and dynamic parameters (e.g., blood flow velocity, using appropriate software) [8]. Studies published in the last few decades have demonstrated the importance of this innovative technique in the following diseases: diabetes, glaucoma, essential hypertension, psoriasis (dermatology), cardiovascular disease, chronic kidney disease (CKD), and Alzheimer’s disease [9]. One of the papers reviewed concluded that NFC findings (such as reduced capillary density and length, ectatic loops, micro-hemorrhages, tortuosity, dilated capillaries, avascular areas or ramified capillaries, or aneurysms) are not infrequent, even in patients with no underlying CTD. A summary of the included studies with significant differences in NFC findings between controls and patients is provided in [3,9]. These studies conclude that NFC provides key data for the determination of vascular damage in diabetic patients and thus enables the evaluation of disease progression [9], making this technique potentially useful in evaluating microvascular disease. Maldonado and colleagues demonstrated that NFC in patients with diabetes enables the identification of capillaroscopic abnormalities, possibly including a characteristic pattern in this group of patients consisting of cross-linked, tortuous capillaries, ectasias, and avascular zones [10,11]. They also found an association between microangiopathies seen by ophthalmoscopy and the presence of avascular zones and ramified capillaries seen at the nailfold level [10,11].

There have been few studies published on NFC and dietary habits, so in addition to reviewing the utility of NFC measurements, the dietary habit survey was expanded to include microcirculation measurements by other methods. This approach allows for summarizing comments on lifestyle and dietary recommendations for peripheral artery disease (PAD) prevention from the viewpoint of microvascular function and blood flow.

## 2. Normal Capillaroscopy Pattern in Healthy Subjects

A non-systematic narrative review of the literature in English was conducted using the PubMed database with the aim of undertaking a comprehensive review on the usefulness of NFC, not only in clinical diagnosis, but also in improving unhealthy dietary lifestyles. Emphasis was on the latter category because, to date, there have been few papers published in the ‘healthy subject’ category. The search was developed with the following terms and descriptors: “nailfold capillaroscopy” [Title/Abstract] AND “Humans” [Mesh] AND “last 43 years” [PDat] OR “last 24 years” [PDat]. This search retrieved 327 papers over the last 43 years across all research fields and these were listed in an Excel file. Most papers retrieved were concerned with ‘non-healthy’ subjects. Through inspection of the titles and abstracts for papers from the last 25 years (1999 to 2023), English-written papers with ‘healthy subjects’ were classified according to whether or not they were related to dietary (or nutritional) lifestyle.

A total of 243 original and 84 review articles were identified from 1999 to 2023. Among these, only 14 were concerned with the lifestyle habits of non-patients, among which those concerned with pregnancy were ignored, for a total of only 8 original papers (10 minus 2 pregnancy papers) and no review papers (4 minus 4 pregnancy papers). These papers included not only “randomized controlled trial studies” but also other small-scale pilot studies. Finally, 7 [12,13,14,15,16,17,18] of the 8 original papers were chosen as suitable original articles to cite in the present review, with one sleep quality and duration study excluded [19]. The survey results of recently published NFC papers (1999 to 2023) are summarized in Figure 2.

Dima, A, et al. summarized qualitative NFC assessment in healthy subjects as follows: Most frequently, capillaries stood perpendicular to the nail edge [3,12,20]. The terminal row of capillaries lay parallel to the surface and enabled many features of the capillaries to be observed [12], including their U-shaped ending, which were clearly visible [21]. Under the rows of capillaries, the subpapillary venous plexus could be seen and appeared larger in diameter [22]. The arterial afferent loop is thinner than the venous efferent loop [20]. As well as the hairpin shape, there were two discrete abnormalities and nonspecific variations that were recognized in the normal nailfold: tortuous capillaries (where the loops meandered but did not cross) and crossing capillaries (where the loops crossed once or twice) [1]. Except for these three characteristics (hairpin, tortuous, crossing), all other morphologies were considered abnormal [1]. Ingegnoli, F., et al. defined the morphological characters of the capillaries as hairpin-shaped loops, loops with one cross, loops with more than two intersections, and meandering loops, and proposed three main capillaroscopic patterns for normal variations found in healthy subjects: (1) “Normal” with 2–5 U-shaped loops/mm and a maximum of 2 tortuous loops/mm, (2) “Perfect Normal”, consisting of 5 or more U-shaped loops–/mm, and (3) “Unusual Normal”. The latter was defined as consisting of at least one of the following: one meandering, one bushy loop, one microhemorrhage, or more than 4 crossed loops/mm [23]. Therefore, unusual capillaries might have occurred even in subjects without underlying scleroderma spectrum disorders (SSD) [22]. Counting the number of capillaries per mm was important for defining normality [23]. Different authors have defined various limits for normal capillary density, including 6–14 capillaries/mm [22], 7–10 capillaries/mm [23], 7–12 capillaries [24], 9–13 capillaries/mm [25], and 9–14 (mean 10) capillaries/mm [5]. A recent standardization by the EULAR Study Group for Microcirculation in Rheumatic Diseases defined normal density as 7 or more capillaries/mm [1] (see Table 1).

Similarly, definitions of capillary loop diameter and length varied. Ectasia was initially defined as capillaries with a diameter 4–9 times the normal size, and giant capillaries as more than 10 times larger [20]. Afferent branches were considered to have a normal diameter range of 6–19 µm, and efferent branches of 8–20 µm, with diameters of more than 50 µm defining mega or giant capillaries [20,22,25]. By consensus, it was recently established that a normal capillary diameter should have a maximum diameter of 20 µm and 50 µm for ectasia [1]. Loop length is the visible part of a capillary and were reported by some authors to range from 200–250 µm [22], and reported by others to extend up to 500 µm [23,25]. Elongated capillaries had a length of more than 300 µm, 500 µm, or even 700 µm [26]. Microhemorrhages might have been seen in healthy individuals in cases of local trauma, in which their position was closer to the nail and not related to a recently collapsed giant capillary [20]. The characteristics of NFC morphology in healthy subjects are summarized in Table 2 [12,21,23,25].

## 3. Nailfold Capillary Patterns Correlate with Age, Gender, Lifestyle Habits

Poor lifestyle and dietary habits can cause cardiovascular diseases [29], cancer [30], diabetes [31], Alzheimer’s disease [32], and other lifestyle-related diseases. A healthy lifestyle, including moderate exercise and reducing psychological stress, smoking, and alcoholic intake, can prevent such diseases. For routine medical examinations, simple, inexpensive, non-invasive methods are preferable. Morphological changes in nailfold capillaries have been reported in patients with unhealthy lifestyle habits such as smoking [13], sleep deprivation, and even psychological stress, all of which can lead to slow blood flow [33]. Therefore, studying the relationships between the morphology of nailfold capillaries and lifestyle habits is likely to provide an indication of an unhealthy state, or even predict the pre-disease condition [14,15]. Several studies have described the normality or normal range of nailfold capillaries in healthy individuals in terms of nailfold capillary morphology [14,15,16,17,23]. Below, four out of seven “non-patient” papers relating to dietary habits, are considered in more detail.

### 3.1. Example 1: Nailfold Capillaroscopy (NFC) of Healthy Individuals—An Observational Study [14], Table 3

The morphology and capillary density of the nailfold capillaries of healthy individuals were studied and analyzed. It was concluded that a number of capillary morphological features (such as tortuous, dilated, meandering, or bushy capillaries) and decreased plexus visibility can be present in non-clinical individuals with a BMI greater than 25 that was statistically significant (*p* = 0.0222, in “Plexus” capillary visibility). “Tortuous” capillaries (*p* = 0.0002) and “Receding” capillaries (*p* = 0.0229) were more frequently seen in the group of individuals aged more than 41 years (Table 3). They also suggested that these characteristics should be present in the nailfold of three fingers or more for them to be considered pathological.

**Table 3 nutrients-16-01914-t003:** Morphological characteristics of nailfold capillaries present in healthy individuals. (Typical example of NFC, from [14]).

Nailfold Capillary	Dilated	Neoangiogenesis	Meandering	Tortuous	Ramified	Plexus Visibility	Micro Hemorrhage	Receding ***	Angulated ****
Male (72)	25	29	34	29	4	29	4	23	9
%	34.72	40.27	47.22	40.27	5.55	40.27	5.55	31.94	12.5
Female (78)	25	29	33	39	6	26	4	24	13
%	32.05	37.17	42.3	50	7.69	33.33	5.12	30.76	16.66
*p*	0.8624	0.8247	0.549	0.3026	0.8442	0.3797	0.9074	0.8768	0.6244
BMI * < 24.9 (84)	30	38	42	36	5	38	4	24	15
%	35.71	45.23	50	42.85	5.95	45.23	4.76	28.571	17.85
BMI * > 25 (66)	20	20	25	32	5	17	4	23	7
%	30.3	30.3	37.87	48.48	7.57	25.75	6.06	34.84	10.6
*p*	0.6007	0.0909	0.1879	0.6016	0.9474	0.0222 **	0.7253	0.5187	0.3108
20–40 years (78)	22	31	34	23	3	32	5	17	11
%	28.2	39.74	43.58	29.48	3.84	41.02	6.41	21.79	14.1
41–60 years (72)	28	27	33	45	7	23	3	30	11
%	38.88	37.5	45.83	62.5	9.72	31.94	4.61	41.66	15.27
*p*	0.225	0.9092	0.911	0.0002 **	0.2654	0.3254	0.8074	0.0229 **	0.9699

* BMI = Body Mass Index, ** significant difference, *** to become less clear or less bright, **** having angles or an angular shape. From Gorasiya, AR, et al. [14].

### 3.2. Example 2: Application of NFC to the Lifestyle Management [15]

Nakajima, T., et al. evaluated the relationship between nailfold capillary morphology, lifestyle habits (particularly those with cardiovascular phenotypes), and coldness of the fingertips, which may indicate a microcirculation disorder suggestive of a pre-disease condition. Their study was undertaken to evaluate the correlation between the morphology of nailfold capillaries and (I) lifestyle habits, (II) clinical experimental design for a broad study, and (III) a follow-up study. In the broad study, both men and women participated and data were collected as fingertip temperatures, images using NFC, and questionnaires about lifestyle habits. In the follow-up study, women participants addressed their improvement in lifestyle habits after the first test, and data were re-analyzed after 1–2 weeks.

Nailfold capillaries were observed under a light microscope (Kekkan-Bijin, AT Co., Ltd., Osaka, Japan; region of observation 500 μm × 700 μm; magnification 320×) after the application of mineral oil to reduce light reflection. Capillary images were captured using a capillary analysis system (CAS; At Co., Ltd., Osaka, Japan). The images of the nailfold capillaries were numerically analyzed for four parameters: diameter, width, length, and distance between the two capillaries. The mean value of these parameters for each individual was used for statistical analysis. The U-shaped area and sums of capillary lengths were measured using a CAS. Representative images of nailfold capillaries showing straight and long loops, twisted or bushy loops, and non-dense and small loops are shown in Figure 3.

They summarized significantly correlated parameters of nailfold capillary morphology and lifestyle habits included in the same group using principal component analysis. In particular, capillary area was negatively correlated with the coldness of fingertips in men aged 20–39 years. Also, capillary width was negatively correlated with sleep problems in men aged 20–39 years, positively correlated with the frequency of smoking in the same group, and negatively correlated with intense exercise in men aged 40 years and over (Figure 4, Lower). In women aged 20–39 years, capillary length was negatively correlated with the frequency of drinking alcohol. Figure 4 (Upper) shows a dot plot indicating a tendency towards smaller lengths of capillary loops in the groups scoring higher for drinking alcohol.

The study by Nakajima, T., et al. could be the first report indicating a correlation between the morphology of nailfold capillaries and lifestyle habits in a non-clinical population involving cardiovascular phenotypes. Therefore, they concluded that the simple, inexpensive, and non-invasive method using NFC can be employed for routine medical examinations everywhere, even at the bedside.

### 3.3. Example 3: LPS Supplement Improved Capillary Vessel and Blood HbA1c Level [16]

An endotoxin is a lipopolysaccharide on the surface of Gram-negative bacteria causing serious inflammation leading to septicemia. However, LPSs are common in the environment and are not toxic when administered orally or trans-dermally, but they contribute to the activation of innate immunity [34]. In 2017, Nakata, Y., et al. [16] used a dietary supplement containing LPS in a randomized double-blind, parallel-group trial study. The LPS was derived from the Gram-negative bacterium, *Pantoea agglomerans* (LPSp; containing 201.5 μg/tablet as LPS), and nailfold capillaroscopy measurements were applied to judge the effectiveness of this dietary supplement.

LPS is present in the environment, attaching to edible plants or floating in the air, and it has been recently revealed that the human immune system is influenced by ingesting this environmental LPS, either via diet or the respiratory tract [35]. Since LPS has been found in Chinese medicines and edible plants, Inagawa et al. [36] focused on the physiological actions of naturally ingested LPS and demonstrated its safety and beneficial effects through oral and transdermal administration.

One tablet of this LPSp supplement was composed of 25 mg of fermented wheat extract containing 10 mg/g of LPSp (measured by ELISA (enzyme-linked immunosorbent assay) as 201.5 ± 22.8 μg/tablet). Fifty-two subjects were used in this study including males and females aged 20 to 74. Before the start of the study, all were within the normal range of hematological values (excluding oxidized LDL, IgA, and CRP; mild abnormalities in the “Japan Society of Ningen Dock Criteria”). In the control group (*n* = 26), the LPSp supplement was replaced with dextrin. They used a CAS (At Co., Ltd., Osaka, Japan) for vascular observation. The capillary vessel and bloodstream in the subungual space of the left ring finger were analyzed microscopically using CAS, and a significant increase was found in the LPSp-supplemented group after three months compared to the control group (Table 4). According to the Japan Diabetes Society (JDS) criteria (https://www.jds.or.jp/modules/en/index.php?content_id=44 (accessed on 19 March 2024)), 24 subjects in the LPSp supplemented group and 20 subjects in the control group were within the borderline range for HbA1c (a glycation marker) at baseline. Including the subjects within the borderline range, a significant decrease in blood HbA1c was found in the LPSp-supplemented group three months after ingestion, indicating an increase in the number of capillary vessels and antiglycative effects associated with oral ingestion of LPSp in healthy subjects.

### 3.4. Example 4: Fermented Herbal Decoction Improves Peripheral Capillary Morphology [17]

Another example of the effects of dietary factors on nailfold capillary morphology was reported using a fermented herbal decoction by Akazawa-Kudoh, S., et al. in 2018. Reduction in the performance status of the skin condition is one of the serious factors determining the prognosis of post-menopausal women. The purpose of their study was to investigate whether a fermented herbal decoction (FHD) could improve the performance status of post-menopausal skin conditions, by inspecting the capillary conditions in the nailfold in the ring finger of the left hand. A commercially available assortment of 80 wild herbs was prepared and extracted with 100 mL of hot water (98 °C) for 3 min and then roasted dry to obtain 10 g of powder. The resulting extract powder was then fermented for 5 days at 40 °C with *Lactobacillus leuteria* using a ratio of 100:50:850 of powdered sample/lactobacilli/water (prepared by Echigo Yakusou, Ltd., Niigata, Japan). After centrifugation of the fermentation product at 2000× *g* for 10 min at room temperature, the resulting supernatant was served as the FHD. A randomized semi-clinical trial was conducted to assess the nailfold capillary length and its age-related value before and after (6 and 12 months) the administration of FHD, which contained **γ**-amino butyric acid (GABA) produced as part of the fermentation mentioned above. FHD (administered either orally or applied to the skin) improved the performance status of volunteers significantly (*p* < 0.01), increasing the regeneration of nailfold capillaries (an increase in the number of capillaries reconstituted in the 12-month group). Other dietary supplement-relating papers like Ref. [37] written in Japanese language were excluded from this section.

## 4. Nailfold Capillaroscopy (NFC) and Its Application for Peripheral Artery Diagnosis

There have been few papers published on NFC and dietary habits, so the dietary habit survey was expanded to include microcirculation measured by other methods and applied to networks of small blood vessels (small arteries, arterioles, capillaries, venules, and small veins). These networks establish a frontier with the interstitium and lymphatic vessels, which aids in maintaining the homeostasis of tissues and the cardiovascular system [38,39,40]. Considered a tissue interstitium, the microcirculation also contributes to the local tissue immune system by allowing the infiltration and adhesion of immune cells [41]. Also, given the small caliber of its components, the microcirculation is a favorable site for the accumulation of hemostatic plugs. From this viewpoint, a summary of the parallel relationship between abnormal NFC data and PAD-related conditions is provided in Table 5.

### 4.1. Example 1: The Effect of Vitamin D Receptor Activator Treatment on Capillary Blood Velocity (CBV) in Chronic Kidney Disease (CKD) Patients [42]

This paper was included in the present review despite dealing with non-healthy patients with moderate CKD and pre-disease control subjects because of the effectiveness of nutrients on this disease. Previously, Dreyer, G., et al. [46] used ergocalciferol (vitamin D2) in a double-blind randomized trial and showed that vitamin D supplementation improved microvascular endothelial function in the skin, as assessed via laser Doppler flowmetry after iontophoresis of acetylcholine in patients with CKD stages 3–4. Chitalia, N., et al. [47] using cholecalciferol and Zoccali, C., et al. [48] using paricalcitol (a vitamin D receptor activator; VDRA) showed similar results on macrovascular endothelial function, as assessed via determining flow-mediated vasodilatation (FMD) in the brachial artery. The following is a summary of the study by Lundwall, K., et al. [42], who used a double-blind placebo-controlled randomized trial, aiming to investigate whether low- or high-dose treatment with VDRA can ameliorate sympathetic activation and macro- and microvascular functions, as assessed by several state-of-the-art methods in non-diabetic patients with moderate CKD. There was borderline significance (*p* = 0.05, [16]) for improved capillary blood velocity (CBV), as measured by NFC of the hallux in the treated group 3 months after receiving 2 μg of paricalcitol.

### 4.2. Example 2: Dynamic Nailfold Videocapillaroscopy May Be Used for Early Microvascular Dysfunction in Obesity [43]

It has been proposed that obesity is the primary cause of microvascular dysfunction, which could be a pathway to increasing blood pressure and decreasing insulin sensitivity. Due to the high global prevalence of this metabolic disorder, Maranhao, et al. investigated which was the most appropriate videocapillaroscopic method (nailfold or dorsal finger) to assess microvascular function in obese patients, since both techniques can be used for both early detection and follow-up. Their results strongly suggest that microvascular dysfunction consequent to obesity could be better detected by dynamic nailfold videocapillaroscopy than by dorsal finger videocapillaroscopy. They speculated that the derangement of microvascular hemodynamics occurs before the diagnosis of hypertension, diabetes, and other metabolic syndromes. Therefore, they advocated that NFC is the most appropriate technique to assess obesity-related microvascular dysfunction as a pre-disease condition.

### 4.3. Example 3: The Relationship between Nailfold Microcirculation and Retinal Microcirculation in Healthy Subjects [18]

Tian, J., et al. selected 50 subjects without systematic or ocular diseases. The thickness of the peripapillary retinal nerve fiber layer (RNFL), vessel density (VD) of the radial peripapillary capillaries (RPCs), and superficial capillary VD in the macular zone were measured via optical coherence tomography angiography (OCTA) of the left eye. Nailfold microcirculation (including capillary density, avascular zones, dilated capillaries, and hemorrhages) was examined on the fourth digit of each subject’s non-dominant (left) hand via NFC. Lower density nailfold capillaries and abnormalities were found to be associated with reduced RNFL thickness and retinal VD. These results provide a basis for relevant studies on the pathogenesis of ocular diseases with microvascular abnormalities. NFC and OCTA therefore have the potential to identify risk factors and improve the accuracy of early diagnosis and treatment of ocular diseases.

### 4.4. Example 4: NFC and Peripheral Artery Diseases (PAD) [44]

Studies on NFC patterns and their related predictive value in patients with PAD were rare before 2022. NFC patterns in patients with PAD are likely to be aberrant, so it is suggested that NFC is applicable for assessing patients with PAD in outpatient clinics where NFC patterns are abnormal. Wijnand, J.G.J., et al. recently collected data prospectively on three randomly selected days from patients visiting the outpatient clinic of a single vascular surgery unit for three months in 2018. Eligible patients consenting to the procedure were those with a history of PAD (intermittent claudication, IC; *n* = 17) or chronic limb-threatening ischemia (CLTI; *n* = 9). Controls without a history of PAD or CLTI (*n* = 10) were visitors accompanying the patients.

The results revealed no significant abnormalities in quantitative measures; the mean capillary diameter was within the normal reference value range [21], and the mean capillary count was similar among healthy controls, IC, and CLTI groups. However, among the qualitative measures, the IC group showed prevalence in both the presence of hemorrhages and non-specific qualitative capillary abnormalities (including disturbed architecture, lower density, abnormal morphology, dilatation, areas of decreased vascularity, and atypical branching). Two observations occurred exclusively in CLTI patients, prominent venous plexus (PVP) and mega-capillaries (MCs). These were parallelled by biomarkers of endothelial dysfunction [49]. In systemic sclerosis, the extent of abnormalities revealed by NFC has been linked to circulating biomarkers of inflammation and endothelial dysfunction [50], which have also been shown to be altered in CLTI. These findings suggest that NFC abnormalities may also be used as markers for inflammation and endothelial dysfunction in PAD, although (as both groups noted) confirmation in larger studies is required.

### 4.5. Example 5: Relationship between NFC Parameters and the Severity of Diabetic Retinopathy [45]

Recently Okabe, et al. investigated whether or not non-invasive measurements with NFC are associated with the presence and severity of diabetic retinopathy (DR) in patients with type 2 diabetes. There were 83 patients with type 2 diabetes and 63 age-matched non-diabetic control subjects. Diabetic patients were classified by the severity of their DR: non-DR (NDR), non-proliferative DR (NPDR), or proliferative DR (PDR). Okabe et al. showed that four NFC parameters in the diabetic patients were significantly lower than in the controls (all *p* < 0.001); that is, there was a statistically significant decrease in the NFC parameters along with the increasing severity of DR (number: *p* = 0.02; all others: *p* < 0.001). Logistic regression analysis revealed that combining the systemic characteristics of age, sex, systolic blood pressure, estimated glomerular filtration rate, Hb A1c level, and history of hypertension or dyslipidemia could indicate the presence of DR or PDR (the area under the receiver operating characteristic curve [AUC] = 0.81, *p* = 0.006; AUC = 0.87, *p* = 0.001, respectively). Furthermore, the discriminative power of DR was significantly improved (*p* = 0.03) by adding NFC length to the systemic findings (AUC = 0.89, *p* < 0.001). Finally, they concluded that alterations in NFC morphology, such as capillary shortening, may be closely correlated with the presence of DR (diabetic retinopathy) or PDR (proliferative DR).

## 5. The Role of Dietary or Nutritional Supplementation in Microcirculation

In this final section, the focus is on the role of dietary or nutritional supplementation in microcirculation as summarized from recently published papers, mainly from studies in the USA and the EU where systems for government approval of dietary supplements preceded those of other countries [51].

### 5.1. Health Claims in the USA and Europe [38]

Food supplements are manufactured from food substances, isolated nutrients, or food-derived compounds, and are available in the form of powders, pills, potions, and other types of medication that are not commonly associated directly with food [52]. Due to the increasing interest shown in personal health, aging demographics, and successful personalized care products, the demand for food supplements has grown and is expected to continue to grow [53]. Cardiovascular disease (CVD) is a significant public health concern worldwide and a leading cause of morbidity and mortality in developed countries [54,55]. In 2015, nearly one-third of all deaths worldwide were caused by CVD according to the World Health Organization [56]. Therefore, the effects of most food supplements on cardiovascular risk and CVD have long since been investigated [53,57,58,59].

### 5.2. American and European Legal Frameworks for Food Supplements

There are different ways of classifying the claims for the recognition of food supplements in the USA and the EU. In the USA, there are three major categories, namely “nutrition content claims”, “structure/function claims”, and “health claims” [60]. In the EU, the three major categories are “nutrition claims”, “health claims”, and “reduction of disease risk claims” [61]. There is neither a consensus nor overlap between these two different classifications. Indeed, from the scope of the review by Raposo, A., et al. [38], it was deemed adequate to consider only two types of claims, “nutritional claims” (which correspond to the EU “nutrition claims” plus the “nutrition content claims” and “structure/function claims” in the USA) and “health claims” (which encompass the USA “health claims” and “health claims” plus the “reduction of disease risk claims” in the EU).

Authorized health claims approved by the US Food and Drug Administration (FDA) must comply with the Significant Scientific Agreement (SSA) health claims or the FDA Modernization Act (FDAMA) health claims. However, only SSA health claims are allowed on food supplement labels [62]. When a food supplement does not fully satisfy the SSA but is still recognized through some scientific evidence that can support its intended claim, the FDA may recognize that claim as a “qualified health claim”. Thus, it is worth knowing which food supplements may have beneficial effects on the microcirculation by considering that the health claims approved by the USA, the EU, or other governments positively affect consumer choices. Although no health claims were identified directly with the word microcirculation in the list of claims approved by the FDA [62], some of these claims may, in fact, have direct and indirect implications for microcirculation (Table 6). These “Authorized Health Claims” are shown on the left side of this table and “Qualified Health Claims” are shown on the right side of this table.

Regarding the EU [63,64], the list of non-authorized claims with the word ‘microcirculation’ are shown in Table 7, which can be generically considered similar to those recognized by the US FDA. There were five claims with specific references to the word ‘microcirculation’ that were not authorized given the EFSA’s previous assessment [65,66,67,68,69]. This list contained “Dry isoflavones soy extract”, “Niacin”, “Bioflavonoids”, “Vitamin E acetate”, and oligomeric procyanidins. It was described that “Bioflavonoids” have a positive effect on microcirculatory tropism by favoring processes that protect small venous vessels; they protect the body from the harmful action of free radicals and the skin from ultraviolet rays [67]. It was also described that “Vitamin E acetate” supports microcirculation and scalp oxygenation [68].

### 5.3. Dietary and Lifestyle Recommendation for Peripheral Artery Disease (PAD) Prevention [70] = 2020

PAD is defined as partial or complete stenosis of ≥1 peripheral arteries ([71,72], 2008 and 2015). PAD affects 3–10% of the Western population and if it remains untreated can have devastating consequences for patients and their families. Researchers in Greece, the USA, the UK, and the Netherlands published a review paper entitled “Nutrition, dietary habits, and weight management to prevent and treat patients with PAD” in 2020 [70]. They analyzed how healthy dietary habits can decrease PAD rates when encouraged in the general population, limiting their findings to peripheral arterial disease but excluding patients with aneurysms and arterial diseases also affecting other organs [72]. PAD prevalence ranges between 3–10% in the general population, but can be as high as 15–20% among the elderly [73]. More than 65% of adults with PAD are overweight or obese, while 78% are characterized by deficiencies in vitamins and minerals [74]. Both suboptimal nutritional status and high-fat mass have been associated with the worsening of ambulatory status and vascular health in patients with PAD and claudication [75]. Nutritional advice and weight management are of paramount significance in PAD management [76].

Sagris, M., et al. [70] summarized the dietary and lifestyle recommendations for PAD prevention. They recommended the establishment of regular consumption of various nutrients and fruits, vegetables, and anti-oxidants along with other recommendations for lifestyle management (such as regular physical activity and avoiding smoking). These recommendations include increased consumption of omega-3 fatty acids, various B vitamins (folic acid, B6, B12), fat-soluble vitamins (vitamin A, D, E), vitamin C, and minerals (zinc intake and a sodium upper limit). While further research attempts are anticipated, emphasis on proper nutrition, dietary intervention and weight management should be part of the PAD prevention and treatment as they described [70].

### 5.4. Advancing beyond the “Heart-Healthy Diet” for PAD [73] = 2015

PAD is a burdensome cardiovascular condition that results from chronic inflammatory insults to the arterial vasculature. Key risk factors include age, gender, type 2 diabetes, hypertension, hypercholesterolemia, hyperhomocysteinemia, smoking, lack of physical exercise, and poor diet, the latter three being modifiable in the development and progression of PAD. A growing body of evidence indicates that imbalanced nutrient intake may contribute to the development and progression of PAD. In 2015, Nosova, E.V., et al. summarized nine categories of nutrients, as well as four diets endorsed by the American Heart Association (AHA) and American College of Cardiology (ACC) that may be prescribed to patients with, or at risk of, PAD [73]. They firstly explained the items of the national “Guidelines for Cardiovascular Disease”. The 2013 AHA/ACC composite Task Force guidelines [77] address nutrition-focused risk reduction strategies for cardiovascular disease in a broad context, with the aim of improving public health. The guidelines endorse a “Heart Healthy Lifestyle” and provide a general framework for incorporating healthy nutrition into lifestyle management to improve blood pressure and lipid control. They emphasize broad nutrient categories associated with better cardiovascular outcomes (e.g., fruits and vegetables, whole grains, legumes, etc.). However, an important limitation is that particular nutrients that may benefit patients with advanced cardiovascular disease have not been explicitly identified. The most specific suggestions are embedded in recommendations for lowering low-density lipoprotein cholesterol (LDL-C); the task force advises obtaining a maximum of 6% of total calories from saturated fat and minimizing intake of foods rich in trans-fatty acids.

The nutrients reviewed by their paper include omega-3 polyunsaturated fatty acids (*n*-3 PUFAs), B vitamins, and anti-oxidants. The diet plans described include the DASH (Dietary Approaches to Stop Hypertension) diet, Mediterranean diet, low-fat diet, low carbohydrate diet, Dr. Dean Ornish’s Spectrum^®^ Diet, and Dr. Andrew Weil’s Anti-Inflammatory Diet, although the latter two are not recommended by national guidelines [78]. Considering the “Strength of Evidence” from their paper, *n*-3 PUFAs, fatty fish, and EPA + DHA must be beneficial for primary and secondary prevention of CVD. “Reduce saturated fat intake to 5–6% of total daily caloric intake” and “daily sodium intake restriction to 2.3 g (~6 g/day sodium chloride)” also must be beneficial for the prevention of CVD. “Dietary fiber intake of 14 g/1000 kcal, or 25 g for adult women and 38 g for adult men” are also recommended for CVD prevention. Nosova, E.V., et al. summarized these findings in their paper’s Tables [73] and classified the strength of evidence and level of evidence.

## 6. The Effects of Dietary Flavonoids on Microvascular Health

Over the past two decades, various experimental and epidemiological studies have shown that the consumption of flavonoid-rich foods is associated with a reduced risk of CVD [79,80]. Flavonoids are a large family of over 5000 hydroxylated polyphenolic compounds that carry out important functions in plants, including attracting pollinating insects, combating environmental stresses such as microbial infection (antifungal activity), and regulating cell growth [81]. Their bioavailability and biological activities in the human body appear to be strongly influenced by their chemical nature.

Vascular function is linked with cognition and brain function, with increased cardiovascular health being associated with greater cognitive performance [82,83,84]. Furthermore, many of the risk factors associated with cardiovascular health are also risk factors for cerebrovascular health, such as hypertension, hypercholesterolemia, and diabetes, with CVD itself having been identified as a risk factor for vascular dementia, caused by a reduction in blood flow to the brain [85,86]. As Rees, A., et al. reported, various flavonoids have significant effects on increasing endothelial function and peripheral blood flow, and thereby on decreasing the risk of CVD [87,88,89,90,91,92]. Rees, A., et al. summarized the effect of flavonoids on endothelial function from the viewpoint of flow-mediated dilation (FMD) measurement. Acute or chronic ingestion of dark chocolate flavanols, chocolate polyphenols, cocoa flavanols, apple polyphenols, and black tea catechins were mostly effective in increasing FMD of peripheral arteries [93,94,95,96,97,98,99,100,101,102,103,104,105,106].

### Biological Effects of Epicatechin and Taxifolin [107] = 2021

Bernatova, I. and Liskova, S. emphasized two typical flavonoids: epicatechin and taxifolin. These flavonoids benefit peripheral blood flow and endothelial function. Both are naturally occurring in various fruits, vegetables, and edible tree extracts. Epicatechin is present mainly in green tea, black tea, cacao, and cacao products (cocoa and chocolate). Cacao beans are considered the most abundant source of EC [108,109]. Taxifolin (dihydro-quercetin) can be found in red onions [110], apples [111], tomatoes, sorghum grain, white grapes, strawberries, mulberries [112], acai berries, peanuts, adzuki bean, pine seeds [113], thyme, and citrus fruits [114]. Taxifolin is also present in high concentrations in conifers such as Siberian larch, French maritime bark, Korean red pine, Himalayan cedar, Indian pine, and Chinese yew, from which it can be produced industrially. Both epicatechin and taxifolin are also broadly available in various commercially available food supplements.

The following is a brief summary of examples of the biological effects of (−)-epicatechin as well as (+)-taxifolin and/or (−)-taxifolin demonstrated in experimental models. These effects may be relevant for the treatment of hypertension and viral infections through improvements in peripheral vascular function and anti-inflammatory activity [107]. Reported effects on peripheral vascular function and cardioprotective- and anti-inflammatory effects are summarized in Table 8 [107,115,116,117,118,119,120,121,122,123,124,125,126,127,128,129,130,131,132,133,134,135,136,137,138,139]. References to the cellular and molecular mechanisms underlying the biological effects of epicatechin and taxifolin are listed in Table 9 [107,115,119,123,128,130,140,141,142,143,144,145,146,147,148,149,150,151,152], including effects on reactive oxygen species (ROS) reduction, activation of anti-oxidant enzymes, endothelial NO elevation, and NF-κB reduction.

## 7. Conclusions and Prospects

In the past decade, morphological changes in the nailfold capillary have been reported due to the impacts of unhealthy lifestyles such as smoking, inappropriate diets, sleep deprivation, and psychological stress, which lead to slow blood flow. Therefore, the study of the relationships between the morphology of nailfold capillaries and lifestyle habits has a high potential to indicate an unhealthy state, or even to predict a pre-disease condition. Therefore, simple, inexpensive, and non-invasive methods such as NFC are useful for routine medical examinations. This narrative and comprehensive review has examined the usefulness of NFC in both clinical diagnosis and improving unhealthy dietary lifestyles.

Because there is a parallel relationship between abnormal NFC data and PAD-related conditions, this review included a summary of the health claims from recently published papers in the USA and EU (regions where there is a well-developed government-approved dietary supplement system), to provide an overview of the role of dietary or nutritional supplementation in promoting healthy microvascular blood flow and endothelial function.

A strategy for promoting dietary and lifestyle health was summarized based on NFC and other related measurements, focusing on microvascular blood flow and endothelial function. However, dietary interventional research with flavonoids or other functional nutrients is still necessary to fully understand the mechanisms of action of flavonoids and nutrients in the human body, preferably working with pre-symptomatic subjects using NFC monitoring.

Recent research suggests that not only essential nutrients but also flavonoids and other functional food components can exhibit cardio- and neuro-protective beneficial effects, as demonstrated by improvements in both peripheral blood flow and endothelial function.

## Figures and Tables

**Figure 1 nutrients-16-01914-f001:**
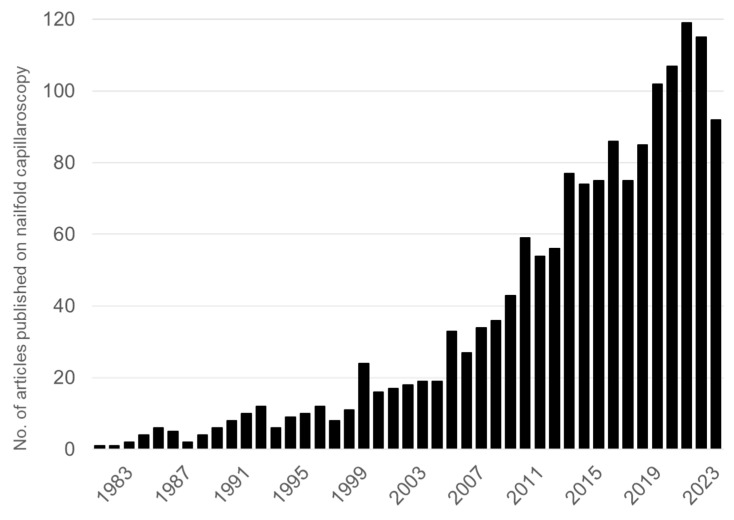
Number of articles published on nailfold capillaroscopy (2023: Up to October).

**Figure 2 nutrients-16-01914-f002:**
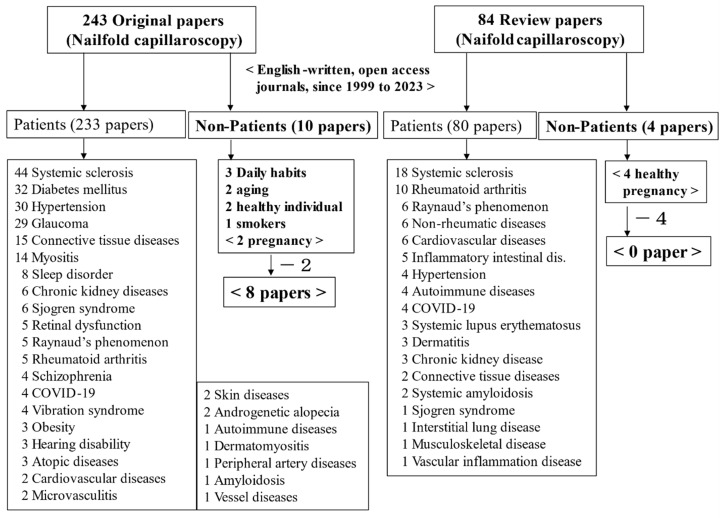
Survey results for recently published papers including the use of nailfold capillaroscopy (1999 to 2023).

**Figure 3 nutrients-16-01914-f003:**
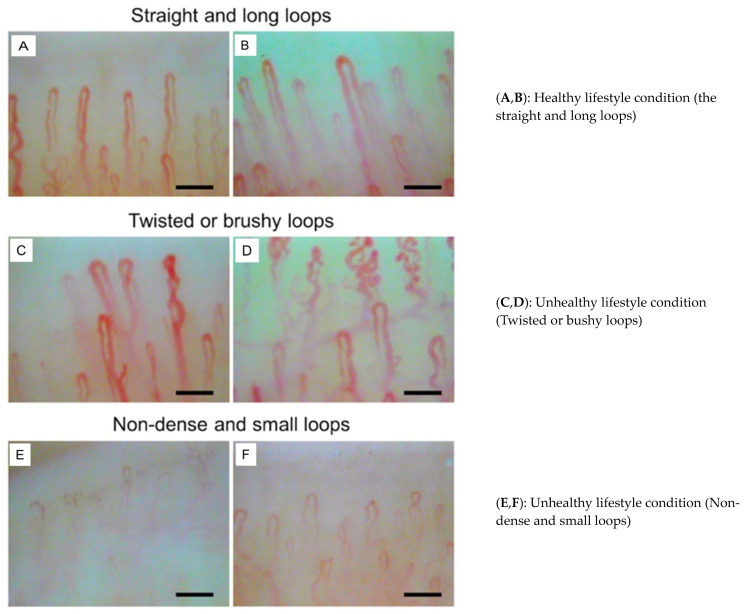
Representative images of nailfold capillaries in healthy subjects [14]. From: Nakajima, T, et al., 2022 [14].

**Figure 4 nutrients-16-01914-f004:**
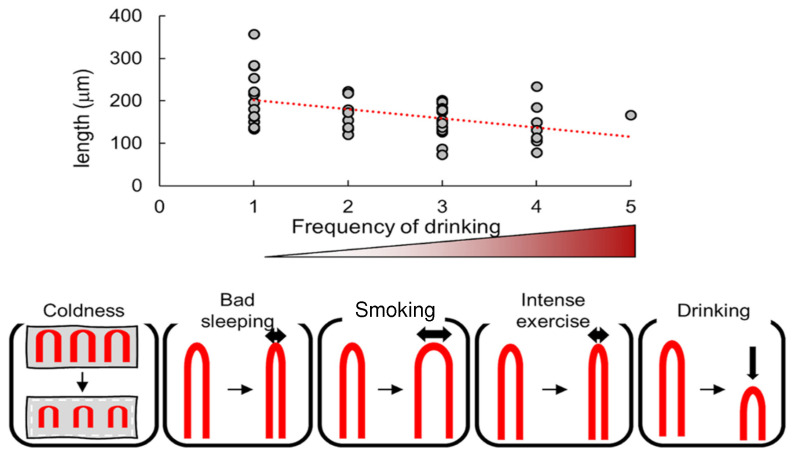
Correlation of nailfold capillary morphology with lifestyle habits (cited from [15]). **Upper**: Dot plot showing loop length of nailfold capillaries negatively correlated with frequency of drinking alcohol. Dotted line: least-squares regression line. **Lower**: Schematic representation of correlation between nailfold capillary morphology and lifestyle habits. Cited from: Nakajima, T, et al. [15].

**Table 1 nutrients-16-01914-t001:** Normal capillaroscopic pattern in healthy subjects. (Summarized from [1,3,4,6,20]).

Parameter	Description
Skin transparency	Allows good visualization of the capillaries
Subpapillary venous plexus	Visible in up to 30% of healthy individuals
General view	Homogeneously sized, regularly arranged
Capillary orientation	Straight, parallel, usually perpendicular to the nailfold
Capillary density	More than 7 capillaries per mm of nailfold
Capillary morphology	Inverted “U”, hairpin shape, but also tortuous and/or crossing capillaries (nonspecific variations)
Capillary length	Less than 300 μm
Capillary diameter	Less than 20 μm for each loop (afferent, apical, efferent)
Pericapillary edema	Absent
Hemorrhages	Absent (occasionally observed after microtrauma)
Giant capillaries	Absent
Neoangiogenesis	Absent
Blood flow characteristics	Dynamic, no stasis

After Smith, V., et al. [1]; Dima, A., et al. [3]; Cutolo, M. [4]; Chojnowski, M.M., et al. [6]; and Kayser, C., et al. [20].

**Table 2 nutrients-16-01914-t002:** Summary of nailfold morphology as observed by capillaroscopy (NFC) in normal healthy subjects [12,21,23,25].

Publication	NFC Morphology
Ingegnoli, F., et al., 2013 [23]	Based on a cluster analysis, three major “normal” morphologic capillaroscopic patterns were recognized: (1) the “normal” pattern, with mainly 2 to 5 U-shaped loops/mm and ≤2 tortuous loops/mm; (2) the “perfect normal” pattern with ≥5 U-shaped loops/mm; and (3) the “unusual normal” with at least 1 meandering or bushy loop, or at least 1 microhemorrhage, or with ≥5 crossed loops/mm. Regarding loop measurements, the majority of subjects had a median of 7 capillaries/mm with a median length of 198 μm.
Faggioli, P., et al., 2015 [25]	Under physiological conditions the normal pattern is characterized by: (1) the orderly arrangement of capillaries to comb; (2) density of 9–13 μm (maximum 3 per dermal papilla); (3) 6–9 µm diameter afferent branch, efferent branch 8–21 µm (>50 µm: megacapillaries); (4) length 200–500 µm.
Tavakol, M.E., et al., 2015 [21]	Nailfold capillary density appears to be similar in healthy adults and healthy children across Europe. European authors found the mean capillary density in healthy children to be in the range of 5–7.3 compared to 7.3–10.3 in healthy adults. Brazilian authors showed slightly higher capillary counts, ranging from 6–7.3 capillaries per millimeter in children and 9.11–10.1 capillaries per millimeter in adults.
Emrani, Z., et al., 2017 [12]	The density of finger capillaries in healthy control subjects were summarized by collecting 17 articles published from 1990 to 2016 as follows. The mean capillary density was 8.45 ± 1.32/mm for individuals aged 40 or less and 8.71 ± 1.40 for individuals older than 40 years of age in healthy subjects *. Mean capillary densities in healthy males and females were found to be 8.83 ± 1.50 and 8.60 ± 1.26/mm, respectively. **

* Ingegnoli, F., Herrick, A.L., 2013 [27]. ** Hoerth, C., et al., 2012 [28].

**Table 4 nutrients-16-01914-t004:** Number of fingertip capillary vessels measured 3 months after ingestion of LPSp supplement [16].

	Control (*n* = 26)		LPSp Supplement (*n* = 26)	
Months (M)	0 M	+3 M	0 M	+3 M
Per field	4.92 ± 0.30	4.42 ± 0.25	4.65 ± 0.25	5.12 ± 0.27 *
Relative value	1.0 ± 0.0	1.057 ± 0.17	1.0 ± 0.0	1.201 ± 0.10

Nailfold capillaroscopy was performed in the area near the lunula of the left ring finger. * Control +3 M vs. LPSp supplement +3 M, *p* < 0.05 by Mann–Whitney U test. From: Nakata, Y., et al. [16].

**Table 5 nutrients-16-01914-t005:** Parallel relationship of nailfold capillaroscopy (NFC) data and peripheral artery disease (PAD)-relating conditions.

Authors and Published Year	Title and Description
<1> Lundwall, K., et al., 2015	Paricalcitol, Microvascular and Endothelial Function in Non-Diabetic Chronic Kidney Disease: A Randomized Trial
[42]	Endothelial function declined significantly over 3 months in patients with moderate CKD, and this decline was ameliorated by vitamin D receptor activator treatment, possibly through increased capillary blood flow.
<2> Maranhao, P.A., et al., 2016	Dynamic Nailfold Videocapillaroscopy may be Used for Early Microvascular Dysfunction in Obesity
[43]	The authors speculate that derangement of microvascular hemodynamics occurs before the presentation of the diagnosis of hypertension, diabetes, or other metabolic syndromes. Therefore, NFC is the most appropriate technique to precociously assess microvascular dysfunction in obesity.
<3> Tian, J., et al., 2020	The Relationship Between Nailfold Microcirculation and Retinal Microcirculation in Healthy Subjects
[18]	There was a direct relationship between nailfold capillary and retinal microcirculation. Therefore, abnormalities seen in NFC are associated with reduced retinal nerve fiber layer thickness and retinal vessel density.
<4> Wijnand, J.G.J., et al., 2022	Naiflold Capillaroscopy in Patients with Peripheral Artery Disease of the Lower Limb (CAPAD Study)
[44]	NFC abnormalities can be used as markers for inflammation and endothelial dysfunction in PAD.
<5> Okabe, T., et al., 2023	Relationship between Nailfold Capillaroscopy Parameters and the Severity of Diabetic Retinopathy
[45]	Alterations in NFC morphology, such as capillary shortening, may be closely correlated with the presence of diabetic retinopathy (DR) and proliferative DR.

**Table 6 nutrients-16-01914-t006:** List of authorized claims in the USA with direct or indirect impact on microcirculation.

Authorized Health Claims	Qualified Health Claims
Dietary saturated fatty acids and cholesterol and risk of coronary heart disease	Whole grain foods with moderate fat content and risk of heart disease
Fruit, vegetables, and grain products that contain fiber *, particularly soluble fiber *, and risk of coronary heart disease	Saturated fatty acids, cholesterol, and trans fatty acids, and reduced risk of heart disease
Soluble fiber * from certain foods and risk of coronary heart disease	Substitution of saturated fatty acids in diet for unsaturated fatty acids and reduced risk of heart disease
Soy protein and risk of coronary heart disease	B vitamins and vascular disease
Plant sterol/stanol esters and risk of coronary heart disease	Nuts and heart disease
	Walnuts and heart disease
Raposo, A., et al., 2021 [38]	Omega 3 fatty acids and coronary heart disease
	Monounsaturated fatty acids from olive oil and coronary heart disease
	Unsaturated fatty acids from canola oil and reduced risk of coronary heart disease
* Fiber = dietary fiber	Corn oil and corn oil-containing products and a reduced risk of heart disease

**Table 7 nutrients-16-01914-t007:** List of nonauthorized claims in the EU with the word microcirculation [38].

Nutrients, Substance, Food, or Food Category	Claim	[Ref] = Year
Dry isoflavones soy extract	Acts on hair bulbs to support hair growth. Prevents hair from premature aging via antioxidant properties and microcirculation.	[65] = 2011
Niacin (B Vitamin)	Activates scalp microcirculation.	[66] = 2009
Bioflavonoids	It has a positive effect on microcirculatory tropism by favoring the processes that protect small venous vessels. It protects the body from the harmful action of free radicals and the skin from ultraviolet rays.	[67] = 2011
Vitamin E acetate (D,L alpha tocopherol acetate)	It supports microcirculation and scalp oxygenation.	[68] = 2010
OPC Plus, containing 40 mg oligomeric procyanidins (OPC) and 40 mg berry blend per capsule	OPC Plus has been shown to increase microcirculation and may, therefore, reduce the risk of chronic venous insufficiency.	[69] = 2020

**Table 8 nutrients-16-01914-t008:** Summary of protective effects of (−)-epicatechin and (+)-taxifolin and/or (−)-taxifolin on cardiovascular diseases in animal models [107].

Biological Effects	(−)-Epicatechin	(+)-Taxifolin and/or (−)-Taxifolin
Vascular	[115,116,117,118,119,120,121,122]	[123]
Cardioprotective	[124,125,126]	[127,128]
Antiinflammatory	[129,130,131]	[132,133]
Antiaggregatory, antithrombotic, or anticoagulant	[115,134,135]	[136,137,138,139]

**Table 9 nutrients-16-01914-t009:** Summary of anti-oxidative and anti-inflammatory effects of (−)-epicatechin and (+)-taxifolin and/or (−)-taxifolin on endothelial function [107].

Cellular and Molecular Mechanisms	(−)-Epicatechin	(+)-Taxifolin and/or (−)-Taxifolin
ROS scavenging	[140,141]	[142]
Activation of antioxidant enzymes (SOD, CAT, GPx)	[143,144]	[128,145]
Elevation of endothelial NO	[115,119,143,146,147]	[123]
NF-kB reduction	[130,148,149]	[150]
Inflammasome reduction	no	[151,152]

ROS: reactive oxygen species; SOD: superoxide dismutase; CAT: catalase; GPx: Glutathione peroxidase; NO: nitric oxide.

## Data Availability

The original contributions presented in the study are included in the article, further inquiries can be directed to the corresponding authors.

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
