# Peer review of "Nailfold Capillaroscopy: A Comprehensive Review on Its Usefulness in Both Clinical Diagnosis and Improving Unhealthy Dietary Lifestyles"

_nutrients, 2024, doi:10.3390/nu16121914_

Round 1

Reviewer 1 Report

Comments and Suggestions for Authors

I’ve read with attention the review by Komai et al. that is interesting, well-organized, overall well-written and update. It is not clear if this review is narrative or systematic, or only partly systematic. I think that the methodology of the review should be further clarified and reported in a dedicated chapter. Then the review should follow a more strict order: NFC and diet/other lifestyle components, NFC and specific dietary supplementation (avoiding examples of strange mixtures tested on a few patients, focusing on specific compounds tested in adequately designed RCTs). 

Comments on the Quality of English Language

No specific concerns. Some typos.

Author Response

To: Reviewer 1:

Thank you for your kind suggestion. 

Re: “It is not clear if this review is narrative or systematic, or only partly systematic.”

<< Answer: We are considering that this review belongs to narrative review rather than systematic, because there have been few published papers (8 papers) on “nailfold capillaroscopy (NFC)” studies in the healthy subject category for 40 years, so we included not only “randomized controlled trial studies” but also small-sized pilot studies. That is the reason why we used “comprehensive” review in the title.  But we are still wondering whether or not it is appropriate to correct the title “comprehensive” review into “narrative” review.  Considering from the Duke University guideline, it may be appropriate to use “narrative” (Duke University Guidance, https://mclibrary.duke.edu)  We would be happy if you could suggest us this thing. 

Re: “the methodology of the review should be further clarified and reported in a dedicated chapter. Then the review should follow a more strict order.”

<< Answer: We modified section 2. (former Lines 95-111 paragraph) as follows. 

  1. Normal capillaroscopy pattern in healthy subjects

A non-systematic narrative review of the literature in English was conducted by utilizing the PubMed database in order to undertake a comprehensive review on the usefulness of NFC not only in clinical diagnosis, but also in unhealthy dietary lifestyle improvements. We concentrated the latter case category because there have been few published papers in healthy subject category up to now. The search was developed with terms and descriptors: “nailfold capillaroscopy” [Title/Abstract] AND “Humans” [Mesh] AND “last 43 years” [PDat] OR “last 24 years” [PDat]. After the survey, we retrieved 327 papers for 43 years belong to all research field and were listed into Excel file, and found out that most of the retrieved papers were belonging to non-healthy patients category. After judging from the papers’ title and abstract for recent 24 years (1999 to 2023), we selected English-written papers with healthy subjects by classifying whether or not relating to dietary or nutritional lifestyle.

Consequently, our search strategy identified 243 original and 84 review articles from 1999 to 2023, and only 8 (10 - 2 pregnancy papers) and no (4 – 4 pregnancy papers) articles were retrieved from the viewpoint of dietary habitual lifestyle in the original and review paper category respectively; even though we cited not only “randomized controlled trial studies” but also other small-sized pilot studies. After all, we chose 7 out of 8 original papers as suitable original articles to cite into the present review because excluded one is sleep-quality and duration paper [12] and not relating to dietary or nutritional one [13, 14, 15, 16, 17, 18, 19]. Therefore, we focused on 7 original papers basically in the present paper to generalize the NFC application for the dietary and nutritional life-style improvement. Surveyed results of recently published NFC papers (1999 to 2023) are summarized in Figure 2.

Reviewer 2 Report

Comments and Suggestions for Authors

The topic of the paper seems to add some value to existing literature in the field of study. Review is very comprehensive. For me isn’t completely clear whether systematic review was undertaken – Authors wrote about systematic search – so if yes, I suggest add such information in the title of article. The Authors presented the results in the healthy subject category (so paper concentrated on patients were not considered). Only 8 such papers were identified, all original (0 reviews). The search strategy assumed reviewing only papers with open-access. Could you justify such restriction? Maybe more papers would be found without such filter. Based on the 8 retrieved papers Authors widely discuss the selected topic. Some disadvantage is that most Tables/plots – are not compilation of information from all 8 original articles prepared by Authors, but rather copies of tables presented in existed literature. Nevertheless, the paper is interesting and worth of attention.

Author Response

To: Reviewer 2

Re: For me isn’t completely clear whether systematic review was undertaken. (Same comment as Reviewer 1)

<< Answer: We are considering that this review belongs to narrative review rather than systematic, because there have been few published papers (8 papers) on “nailfold capillaroscopy (NFC)” studies in the healthy subject category for 40 years, so we included not only “randomized controlled trial studies” but also small-sized pilot studies. That is the reason why we used “comprehensive” review in the title.   But we are still wondering whether or not it is appropriate to correct the title “comprehensive” review into “narrative” review.  Considering from the Duke University guideline, it may be appropriate to use “narrative” (Duke University Guidance, https://mclibrary.duke.edu)  We would be happy if you could suggest us this thing. 

Re: The search strategy assumed reviewing only papers with open-access. Could you justify such restriction? Maybe more papers would be found without such filter. 

<< Answer: Most papers out of 327 were detected in the “open-access” journal in the PubMed database, however, for example there were 10 Japanese-written papers which were not open-access one.  Basic restriction for the present search is English-written papers which have been published in the world, so we can say that there were no more relating papers than 8 original papers in the English-written journals. 

Re: Some disadvantage is that most Tables/plots – are not compilation of information from all 8 original articles prepared by Authors,

<< Answer: We chose 7 out of 8 original papers as suitable articles to cite into the present review because excluded one is sleep-quality and duration paper and not relating to dietary or nutritional one (reference #: 13, 14, 15, 16, 17, 18, 19); and only important Figures and Tables from 4 open-access journals were cited for the presentation.  As a matter of course, we did not change the original papers’ Figure or Table contents basically, however, for readers’ better understanding we modified the contents of Figure 3 and 4 [reference #27] in part because these are co-authors’ paper (Dan Takeno, Chiharu Fujii, and Joe Nakano et al.).  The present review paper contains not only 7 original papers, but also expands the dietary habit survey into microcirculation measurement by other methods because there have been few papers published relating to the NFC and dietary habits.  Therefore, we finally summarized the dietary and lifestyle recommendation for peripheral artery disease (PAD) prevention from the viewpoint of microvascular function and blood flow which might have original idea and priority.

According to reviewers’ question and suggestion, we modified section 2. (former Lines 95-111 paragraph) as follows as a whole.

  1. Normal capillaroscopy pattern in healthy subjects

A non-systematic narrative review of the literature in English was conducted by utilizing the PubMed database in order to undertake a comprehensive review on the usefulness of NFC not only in clinical diagnosis, but also in unhealthy dietary lifestyle improvements. We concentrated the latter case category because there have been few published papers in healthy subject category up to now. The search was developed with terms and descriptors: “nailfold capillaroscopy” [Title/Abstract] AND “Humans” [Mesh] AND “last 43 years” [PDat] OR “last 24 years” [PDat]. After the survey, we retrieved 327 papers for 43 years belong to all research field and were listed into Excel file, and found out that most of the retrieved papers were belonging to non-healthy patients category. After judging from the papers’ title and abstract for recent 24 years (1999 to 2023), we selected English-written papers with healthy subjects by classifying whether or not relating to dietary or nutritional lifestyle.

Consequently, our search strategy identified 243 original and 84 review articles from 1999 to 2023, and only 8 (10 - 2 pregnancy papers) and no (4 – 4 pregnancy papers) articles were retrieved from the viewpoint of dietary habitual lifestyle in the original and review paper category respectively; even though we cited not only “randomized controlled trial studies” but also other small-sized pilot studies. After all, we chose 7 out of 8 original papers as suitable original articles to cite into the present review because excluded one is sleep-quality and duration paper [12] and not relating to dietary or nutritional one [13, 14, 15, 16, 17, 18, 19]. Therefore, we focused on 7 original papers basically in the present paper to generalize the NFC application for the dietary and nutritional life-style improvement. Surveyed results of recently published NFC papers (1999 to 2023) are summarized in Figure 2.
